# DeMBA: a developmental atlas for navigating the mouse brain in space and time

Harry Carey[1,3], Heidi Kleven [1,3], Martin Øvsthus [1], Sharon C. Yates [1], Gergely Csucs [1], Maja A. Puchades [1], Simon McMullan [2], Jan G. Bjaalie [1], Trygve B. Leergaard [1] & Ingvild E. Bjerke [1] ✉

Studies of the adult mouse brain have benefited from three-dimensional (3D) atlases providing standardised frameworks for data analysis and integration. However, few 3D atlases exist for the developing mouse brain, and none cover more than a few ages. This fails to represent the dynamic nature of development and is insufficient for researchers whose data fall between the ages of existent atlases. Existing atlases also neglect the spatial correspondence between ages, making comparison across ages difficult. We present the Developmental Mouse Brain Atlas (DeMBA), a 4D atlas encompassing every postnatal day from 4 to 56. DeMBA was created by using existing brain templates for six developmental time points and interpolating intermediate ages. DeMBA is incorporated in software for spatial registration and analysis, enabling brain-wide quantification of features of interest. Furthermore, we provide the software package CCF translator, allowing coordinates or image volumes to be transformed between ages for comparison. These resources provide age-specific spatial frameworks incorporated in accessible analytic pipelines for developmental mouse brain data.

Three-dimensional (3D) brain atlases and associated digital software are critical for large-scale neuroscientific studies[1,2]. They provide standardised spatial frameworks for analysis, interpretation, and integration of data from diverse sources and are essential resources for studying the structure and function of the brain[3]. 3D atlases are also crucial for more efficient analysis of massive brain-wide datasets[4,5], where they enable data to be interpreted within a common spatial context[6]. An important component of understanding the brain is studying its development, and to this end, many comprehensive datasets have been produced from subjects in the foetal and early post-natal periods[7,8]. Given that the anatomy of the brain changes considerably during development[9], 3D atlases of the developing brain are necessary for accurate integration of these data.

For the mouse, efforts to create 3D atlases have almost exclusively focused on the adult brain[2]. These adult mouse brain atlases have been invaluable – with thousands of datasets mapped to them and shared publicly, they have proven to be a cornerstone of modern neuroscientific practice[10]. Several 3D atlases of the developing brain have recently been established and have already supported multiple discoveries[11–14]. While these atlases have huge utility to the field, a major limitation is that they have all focused on only a subset of discrete developmental ages[13–16]. This approach has notable drawbacks. First, when the temporal dimension of an atlas is reduced to a subset of ages, it is unable to accurately represent data from animals whose age is not in this subset. Second, this approach treats each age as a separate atlas without considering the spatiotemporal relationship between them. While the different atlases may include annotations delineating the same regions, allowing gross comparison (e.g., the number of cells in a particular region[13]), the lack of a connection between them prohibits more advanced analysis. For example, direct voxel-wise comparison of gene expression and co-visualisation of data from different ages are just two of the use cases not currently possible. Thus, current

[1]Neural Systems Laboratory, Institute of Basic Medical Sciences, University of Oslo, Oslo, Norway. [2]Macquarie Medical School, Faculty of Medicine, Health & Human Sciences, Macquarie University, Marsfield, NSW, Australia. [3]These authors contributed equally: Harry Carey, Heidi Kleven. ✉e-mail: i.e.bjerke@medisin.uio.no

approaches will lead to developmental data that are difficult to compare as they are registered to disconnected atlases for different ages. The lack of an atlas incorporating time as a continuous dimension is therefore an impediment to current efforts for large-scale acquisition and integration of data from developing brains.

To overcome these limitations, we created the Developmental Mouse Brain Atlas (DeMBA), a 4D atlas representing age as a continuous variable. DeMBA is generated from six brain image templates (from postnatal mouse brains at ages 4, 7, 14, 21, 28 and 56). We used 3D-to-3D registration to co-register these templates to each other, defining transformation matrices from which we have interpolated models and brain region segmentations for every postnatal day from 4 to 56. Thus, DeMBA establishes a spatial correspondence between each postnatal day. To ensure that this key feature of the atlas, a traversable time dimension, is accessible, we provide the Python package *CCF translator*. This software allows researchers to translate coordinates and image volumes from one age to another, and to use data from two ages to create models of intermediate ages. DeMBA is openly accessible via the EBRAINS Knowledge Graph[17] and is integrated in software for localising, visualising, and quantifying features of interest in serial 2D images[18–21], facilitating its use in a broad range of research applications.

## Results

The Developmental Mouse Brain Atlas (DeMBA) is a four-dimensional (4D) reference atlas covering every postnatal day (P) from 4 to 56 (Fig. 1). It is provided as an open-access resource through the EBRAINS Knowledge Graph[17] and has been integrated in software for spatial registration and analysis (https://www.ebrains.eu/tools/quint-workflow). In the following, we briefly summarise the creation and key features of the atlas, before elaborating on the validation of its anatomical accuracy across ages. We go on to give examples on how to use DeMBA and associated software to study brain development, demonstrating how the atlas and related software can be used for brain-wide analysis of features-of-interest and how coordinates or image volumes can be translated and interpolated across ages.

### Four-dimensional atlas of the developing mouse brain

DeMBA was constructed from serial two-photon tomography (STPT) volumes sourced from public datasets[2,13,14] representing ages P4, P7, P14, P21, P28, and P56, hereafter referred to as 'templates'. Each of these templates is a population average created by combining multiple animals ($N = 6$ animals at P4[14,22], 8 at P7, 15 at P14, 12 at P21, 17 at P28[13], and 1675 at P56[2]). Using the P56 template (the Allen mouse brain Common Coordinate Framework (Allen CCFv3; RRID:SCR_020999; P56))[2] as the starting point, we spatially registered each template to its temporal neighbour, thereby establishing spatial correspondence between each pair. Spatial transformations were based on a three-step (translation, affine, b-spline) registration using elastix[23] (RRID:SCR_009619; https://elastix.lumc.nl/), optimised by manually defined regions and landmarks (for details, see "Methods"). Using the transformation matrices, we then synthesised interpolated models for every postnatal day between the template ages (Fig. 1a and Supplementary video 1). To achieve this, we created a Python package called *CCF Translator* (https://github.com/brainglobe/brainglobe-ccf-translator). *CCF Translator* uses deformation matrices to translate coordinates or image volumes (e.g., template or segmentation volumes) across different coordinate spaces and interpolate between them to create intermediate spaces. Thus, we transformed and interpolated the segmentation volumes from the Allen CCFv3[2] and the Developmental Common Coordinate Framework (a developmental ontology based atlas; DevCCFv001) provided by Kronman and colleagues[15,24] to all ages in DeMBA (P4-P56; Fig. 1b–q).

### Landmark-based validation of the transformations between ages

The anatomical accuracy of the transformations between templates are key to the reliability of DeMBA. During the construction of DeMBA, we assessed this qualitatively by inspecting the results of each age-to-age transformation during the optimisation of parameters and through the choice of manually defined coordinates and regions (for details, see "Methods"). Visual inspection of the segmentations and templates indicated a good fit for major brain regions throughout the brain (Fig. 1c–q).

To quantitatively assess the accuracy of DeMBA transformations, we used a landmark-based approach (Fig. 2). Fifty-four landmarks (Supplementary Data 1) were identified and defined in the P56 template by an experienced neuroanatomist and then independently identified across all template ages (P4, 7, 14, 21, 28) by three other expert neuroanatomists. This allowed us to compare the performance of the DeMBA transformation matrix against the interrater variability of experienced neuroanatomists (hereafter referred to as raters; Fig. 2a). We calculated the mean Euclidean distance of each DeMBA transformation matrix coordinate to the median of the raters' coordinates (Fig. 2a, bar labelled "DeMBA matrix"). This process was then repeated for each rater (Fig. 2a, bars labelled "Rater 1–4") and averaged across them (Fig. 2a, bar labelled "Average rater") to allow statistical comparison. We interpreted this distance as a measure of error for each rater and the DeMBA matrix. The difference between the DeMBA matrix and average rater error was not significant at any age (Fig. 2a1–a5, see also Supplementary Data 2 for all statistical details), validating the DeMBA transformations matrix to be on average as accurate as a neuroanatomist expert. Inspecting heatmap visualisations of the average landmark distances (Fig. 2b) and organising the landmark coordinates by the region in which they were located (Fig. 2c) gives an overview of how the DeMBA matrix distance to the median of raters vary across the brain.

### Using DeMBA and associated software to study brain development

DeMBA is an open access 4D atlas resource for analysing and integrating neuroscience data from the developing mouse brain. The atlas can be cut in any orientation to explore the developing neuroanatomy in standard (Fig. 3a) and arbitrary planes (Fig. 3b). To facilitate the use of the atlas, we implemented it in the suite of software in the QUINT workflow[18–21]. These software enable spatial registration of section images (Fig. 3c–f) and quantification of features-of-interest in histological data with reference to atlas segmentations (Fig. 4). Images spatially registered to DeMBA[25,26] can be viewed with segmentations from the Allen CCF (Fig. 3d–f) and DevCCFv001 (Fig. 3d'–f'). DeMBA can also be used for 3D-to-3D registration (e.g., with the elastix[23] or ANTsX[27] toolkits). The potential of spatially registered data can be leveraged through the *CCF Translator*, which provides the means to transform data and segmentation volumes available for the adult to developmental ages. Our transformation and interpolation of the STPT templates and related segmentation volumes (Allen CCFv3 and DevCCFv001; Fig. 3g–i) exemplifies this use. It can also be used on templates of different modalities that may only be available for selected ages. For example, we transformed and interpolated the light-sheet fluorescence microscopy (LSFM; Fig. 3j) and magnetic resonance imaging (MRI; Fig. 3k) templates provided by Perens and colleagues[5] to all ages in DeMBA[17].

To exemplify the use of the QUINT workflow with DeMBA, we analysed calbindin-stained section images from four age groups (P4, P14, P28 and P56; Fig. 4a–d) available from the Allen Institute in situ hybridisation database[28,29]. The images were linearly and non-linearly registered to age-matched DeMBA templates using QuickNII and VisuAlign, respectively[19,30]. Based on this registration, we exported atlas maps from VisuAlign, using both the Allen CCFv3, 2017 edition

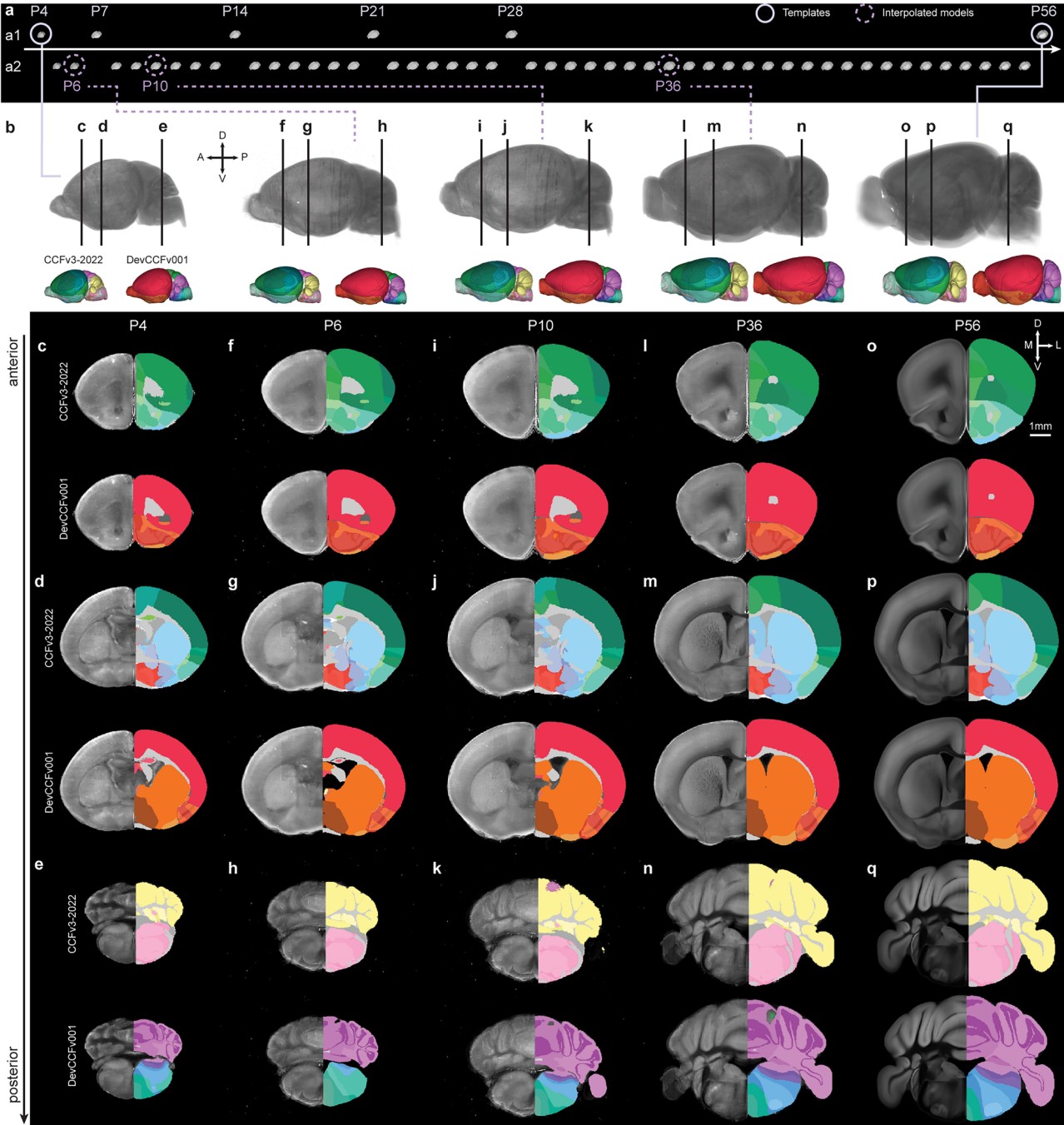

**Fig. 1 | Developmental mouse brain atlas (DeMBA) templates and segmentations with continuous temporal coverage from postnatal day 4–56. a** Template (**a1**) and interpolated model volumes (**a2**) from P4 to P56. **b** Selected template volumes (P4 and P56, full circle) and interpolated models (P6, P10 and P36, dashed circles) with corresponding segmentation volumes from the Allen mouse brain Common Coordinate Framework version 3, 2022 edition (CCFv3-2022) and Developmental Common Coordinate Framework version 001 (DevCCFv001). **c**–**q** Coronal sections at three anteroposterior levels showing the template or model (left side) and the CCFv3-2022 and DevCCFv001 segmentation (upper and lower right side, respectively) for the ages indicated in (**b**). Abbreviations: A anterior, D dorsal, L lateral, M medial, P posterior, V ventral.

(Fig. 4e–h) and DevCCFv001 segmentation (Fig. 4i-l). The atlas maps were combined with segmented images (downloaded from the Allen Institute; Fig. 4a”–d”) using Nutil[21]. This allowed us to quantify differences in calbindin expression across development according to brain region segmentations from the two atlases (Fig. 4m, n). For an analysis of the correspondence of regions in the two atlases, see Kronman et al.[15]. The example analysis presented here shows the use of DeMBA for quantifying features-of-interest in the developing mouse brain with either of the two segmentation sets provided.

The *CCF Translator* also enables researchers to compare data across ages or create interpolated model data for all DeMBA ages. To exemplify this, we established a 4D map of calbindin expression (Fig. 5a, b and Supplementary video 2). This was achieved by first creating 3D heatmaps representing the calbindin neuronal load presented in Fig. 4. Second, data from the four ages (P4, P14, P28 and P56) were interpolated to create models for intermediary ages (see P6 volume in Fig. 5c). Another potential use of the CCF translator is to translate data into the space of any age, for direct comparison of data across ages. For example, the P56 and P4

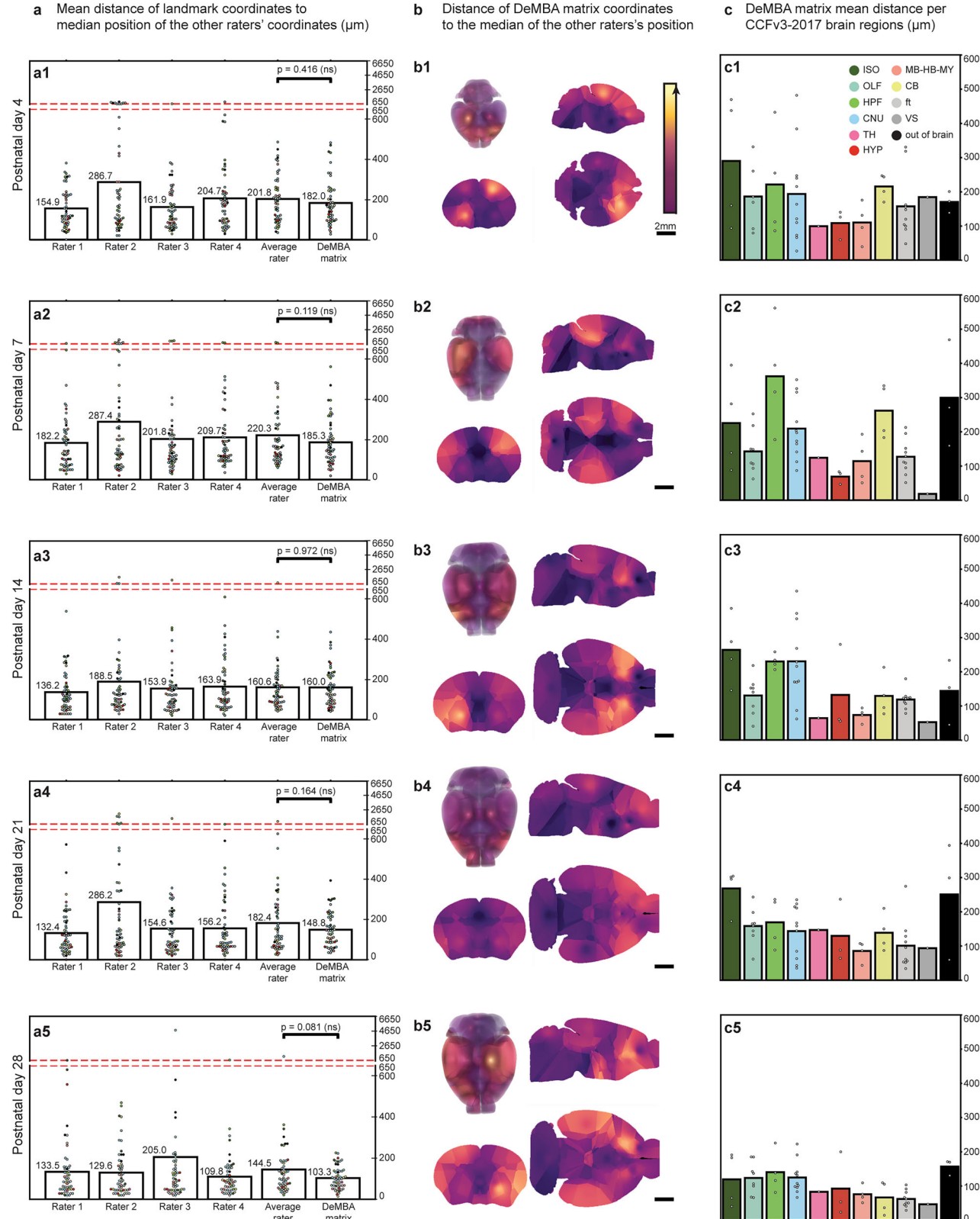

**a** Mean distance of landmark coordinates to median position of the other raters' coordinates (µm)

**b** Distance of DeMBA matrix coordinates to the median of the other raters's position

**c** DeMBA matrix mean distance per CCFv3-2017 brain regions (µm)

calbindin datasets are difficult to directly compare due to substantial differences in neuroanatomy (e.g., in the superior colliculus and cerebellum regions, indicated by red arrowheads in Fig. 5d, e). The adult data can be translated to P4 space for comparing adult and P4 expression in the same space (Fig. 5f). This comparison reveals that calbindin expression in P4 is mainly localised to ventral parts of the cortex and medial and posterior parts of the thalamus, while the adult expression is stronger dorsally (black and white arrowheads in sagittal and coronal plane in Fig. 5d, f). Black and white arrowheads in Fig. 5d–f indicate areas where expression is markedly lower or higher in P4 than in the adult.

## Discussion

DeMBA is a four-dimensional (4D) reference atlas that covers every postnatal day from 4 to 56. We have exemplified how the atlas can be

**Fig. 2 | Accuracy of transformation between template ages. a** Bar plots showing the mean distance of landmark coordinates (*n* = 54 landmarks) as identified by human raters (*n* = 4, "Rater 1–4"), the average rater ("Average rater"), and the DeMBA transformation matrix ("DeMBA matrix") to the median of the other raters' coordinates. The distance is expressed in micrometres (μm). The *y*-axis is truncated from 650 μm upwards due to outlier values. The mean DeMBA error ranged from 103.3 to 185.5 μm, whereas the mean error for the average rater ranged from 144.5 to 220.3 μm across the ages. (**a1**–**a5**) Individual bar plots for each template age (P4, P7, P14, P21 and P28). The average rater and DeMBA matrix distances were compared with a two-tailed *t* test. **b** Heatmap visualisations of the voxel-wise distance

between DeMBA matrix coordinates and the median of raters, with the intensity of the colour indicating the mean error per voxel, i.e., a brighter colour indicates higher errors. Data are interpolated based on the nearest assessed coordinates to provide 3D representations. **c** Mean distance of DeMBA matrix coordinates to the median coordinate of raters organised according to the major brain region in which the coordinate is located (*n* = 54 landmarks). The HPF and ISO generally showed higher error than other areas. Abbreviations: CB cerebellum, CNU cerebral nuclei, ft fibre tracts, HPF hippocampal formation, HYP hypothalamus, ISO isocortex, MB-HBMY midbrain-hindbrain-medulla, OLF olfactory areas, TH thalamus, VS ventricular system.

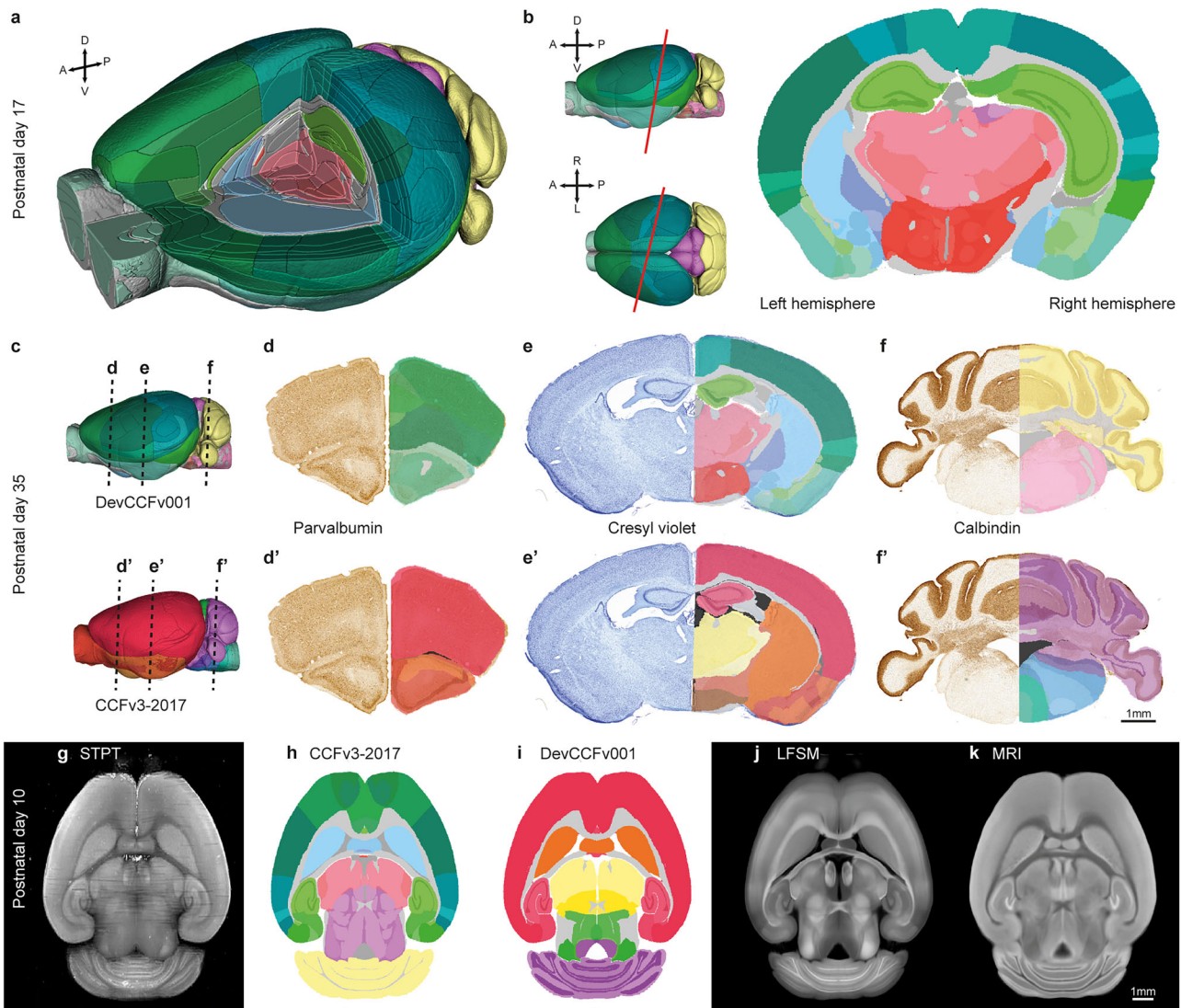

**Fig. 3 | Using DeMBA and associated software. a** 3D surface visualisation of the DeMBA P17 segmentation volume, showing slicing along the standard, orthogonal coronal, sagittal and horizontal planes. **b** 2D surface visualisation of DeMBA, showing slicing along an angle in the dorsoventral and mediolateral plane, resulting in an oblique non-standard plane. **c–f** Three coronal sections stained for parvalbumin (**d**, **d'**), cresyl violet (**e**, **e'**) and calbindin (**f**, **f'**), with semi-transparent overlaid atlas maps from the Allen Common Coordinate Framework version 3 2017 edition (CCFv3-2017) (**d–f**) and the Developmental Common Coordinate Framework

version 001 (DevCCFv001) (**d'–f'**). The sections are from a P35 animal registered to the DeMBA P35 template[25,26]. **g–k** Horizontal sections from the serial-two-photon tomography (STPT) P10 DeMBA template (**g**), and several datasets transformed to the same space using *CCF Translator*: CCFv3-2017 segmentation (**h**), DevCCFv001 segmentation (**i**), light sheet fluorescent microscopy (LSFM) (**j**), and magnetic resonance imaging (MRI) (**k**) datasets. Abbreviations: A anterior, D dorsal, L left, P posterior, R right, V ventral.

used for a range of common analytical purposes, demonstrating its broad practical value for fundamental neuroscience research in the postnatal mouse. In the following, we discuss some of the challenges in creating 3D mouse brain atlases suitable for analysing heterogeneous neuroscience data. We also elaborate on the advances DeMBA offers

for brain development research and for efforts to integrate data across ages and modalities.

Atlases for the developing mouse brain have typically been based on manual segmentation of histology data, yielding 2D atlases[29], which have in some cases been reconstructed to create 3D representations[31]

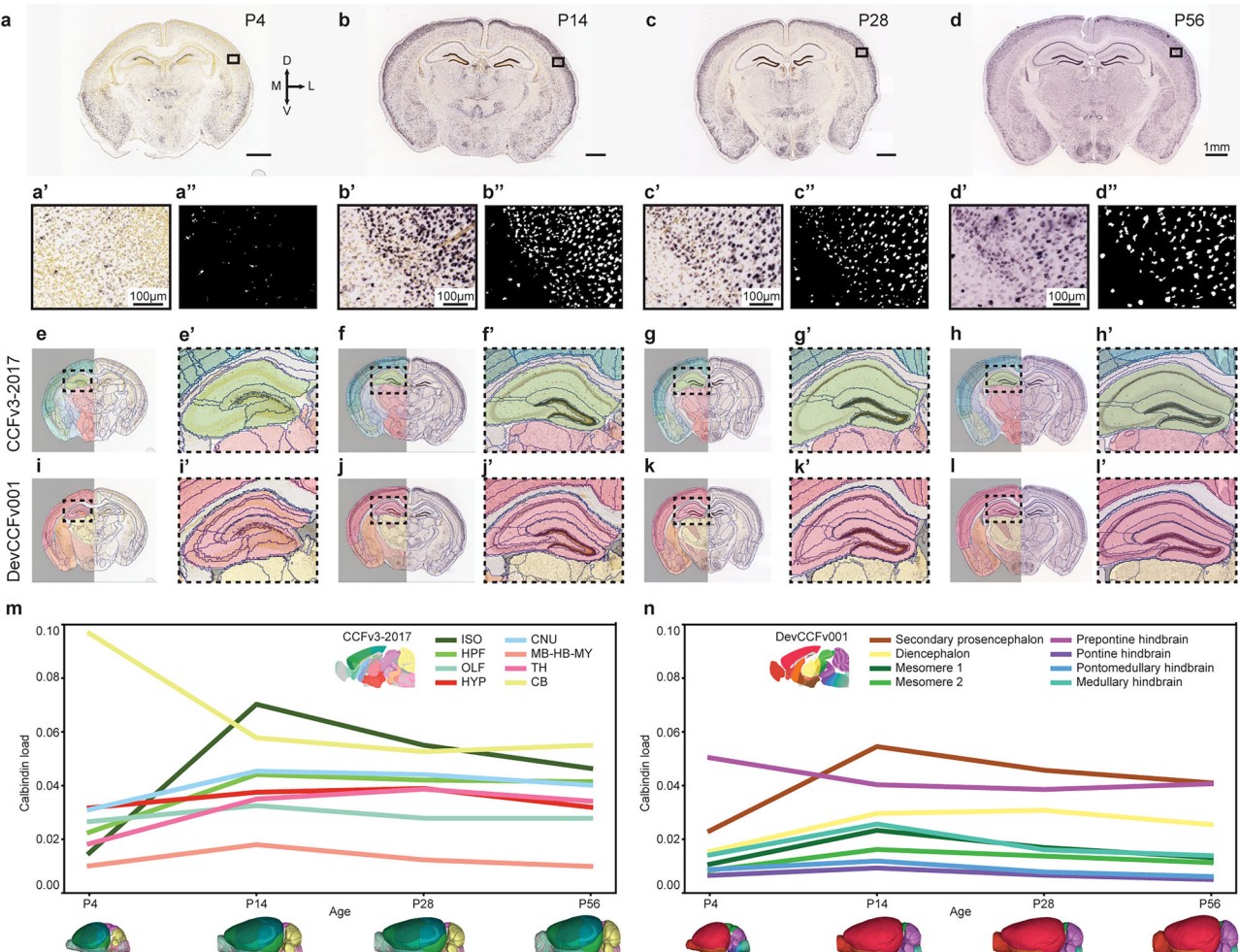

**Fig. 4 | Brain-wide analysis of section images using the QUINT workflow with DeMBA. a–d** Coronal example section images from the Allen Institute in situ hybridisation database, showing calbindin expression at four ages (P4, P14, P28 and P56). The presented analysis includes one brain-wide series per age. **a'–d'** Magnified images from the isocortex. **a''–d''** Segmented images corresponding to (**a'–d'**). **e–l** Atlas maps from the VisuAlign software, corresponding to (**a–d**) used for quantification. **e–h** Segmentation maps derived from the Allen Common Coordinate Framework version 3, 2017 edition (CCFv3-2017) and (**i–l**) the Developmental Common Coordinate Framework version 001 (DevCCFv001) atlas. **e'–l'** Magnified atlas maps, corresponding to the inset in (**e–l**). **m, n** Line graphs showing the calbindin load (area fraction; y-axis) across development (x axis), quantified by use of atlas maps from the CCFv3-2017 (m) and DevCCFv001 (n). Note that the colours in panels m and n are selected based on the colours used in the two atlases. Thus, similar colours in these two graphs do not represent the same regions. Abbreviations: CB cerebellum, CNU cerebral nuclei, D dorsal, ft fibre tracts, HPF hippocampal formation, HYP hypothalamus, ISO isocortex, L lateral, M medial MB-HBMY midbrain-hindbrain-medulla, OLF olfactory areas, TH thalamus, V ventral, VS ventricular system.

(see also, Kronman et al.[15] for an overview of existing developmental atlases). Only a few studies have manually delineated brain regions at several developmental timepoints, most notably Kronman and colleagues[15] who created manual segmentations based on developmental ontology for embryonic and early postnatal stages. However, the effort involved in manual delineation restricts the temporal resolution and variety of template types for which atlases can be made. Using 3D-to-3D registration to fit existing, high-quality segmentation volumes to a wide range of brain templates representing different modalities and ages is a practical alternative to manual segmentation[5,13,14]. Moreover, having identically defined atlas segmentations adapted to age-specific brain morphologies increases the comparative power for studies investigating changes occurring through development. DeMBA builds on recent efforts to use 3D-to-3D registration in creating diverse brain atlases[5,13,14], and provides an example of how the accuracy of spatial transformations can be systematically compared to neuroanatomy expertise. A limitation of DeMBA is that the templates were sourced from multiple independent sources, and thus, there are minor discrepancies between them. For

instance, DeMBA appears to show the brain shrinking between P28 and P56, however, it is likely this is due to methodological differences in the construction of these two templates rather than reflecting genuine morphological changes.

The approach of using 3D-to-3D registration described here also provides the technical base needed to interpolate data between ages. While some studies have provided proof-of-principle in transforming developmental mouse brain data between ages[32] and interpolating between templates[27,33,34], DeMBA and *CCF Translator* provide a solution which covers a broad range of ages, is implemented in software, and has a user-friendly programmatic interface. The practical value of this software is exemplified by the transformation of the adult mouse brain magnetic resonance imaging and light sheet fluorescent microscopy (LSFM) volumes[5] to all the ages available in DeMBA. Such transformed data provide a model of what different modalities would look like for different ages, which we believe can provide a useful resource in the absence of age-specific data. For example, spatial registration of postnatal day (P) 7 LSFM data to a P7 LSFM model might be more convenient than registration to a

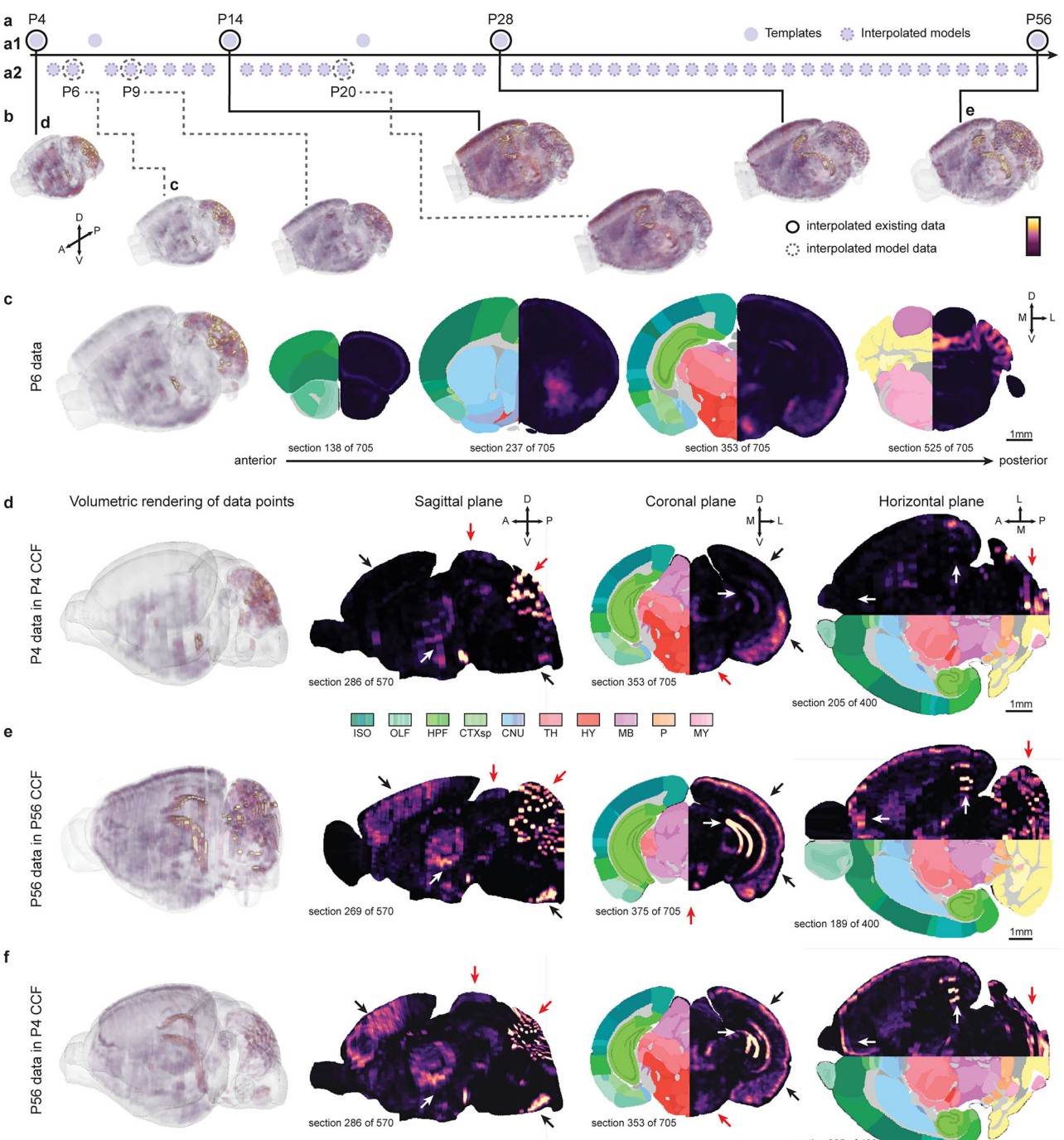

**Fig. 5 | Interpolating and translating developmental data with CCF translator.**
**a** Spheres representing interpolated calbindin data from P4 to P56. The full circles (P4, P14, P28 and P56; **a1**) represent the interpolated data shown in Fig. 3, while dashed circles (P6, P9 and P20; **a2**) indicate interpolated model data for selected ages in between. **b** Seven highlighted volumes of interpolated calbindin data created with the *CCF Translator*. The intensity of the colour indicates the calbindin load per voxel, i.e., a brighter colour indicates a higher load. **c** Interpolated model of the calbindin expression at P6 shown in four coronal sections from anterior to posterior with hemibrain Allen CCFv3, 2022 edition segmentation. **d**–**f** P4 data (**d**), adult data (P56; **e**) and P56 data transformed to P4 space (**f**). Black and white arrowheads indicate areas where expression is markedly lower or higher in the adult than P4, red arrowheads indicate differences in neuroanatomy. Abbreviations: A anterior, CB cerebellum, CNU cerebral nuclei, CTXsp cortical subplate, D dorsal, HB hindbrain, HPF hippocampal formation, HY hypothalamus, ISO isocortex, L lateral, MB midbrain, M medial, MY medulla, OLF olfactory areas, P pons, P posterior, TH thalamus, V ventral.

P7 serial two-photon tomography template, which have a different tissue contrast to the LSFM data. As templates representing different modalities are developed, *CCF Translator* will ensure that they are available for the developing brain. It is also important to note that the method through which *CCF Translator* translates data between ages is not specific to development. In the future, we hope to extend the functionality of *CCF Translator* such that it can transform data not only between ages but between other spaces available for the adult mouse brain, such as the Princeton light sheet space[35] or the Gubra MRI space[5]. We believe that the *CCF Translator* may prove invaluable for the integration of data from both the adult and developing mouse brain.

To summarise, DeMBA is a 4D atlas providing a dynamic representation of the developing brain. It was established based on the spatial transformation of the most widely used common coordinate framework for the adult mouse brain to templates from different postnatal ages. Incorporated in user-friendly software, DeMBA can be used for spatiotemporal analysis and interpolation of templates and data to models of intermediate ages. We envision this atlas and associated software will allow researchers to substantially improve the accuracy and efficiency of developing mouse brain studies. It combines time as a continuous variable with detailed segmentations and user-friendly software for spatial registration and comparative analyses. This provides the neuroscience field with crucial resources for basic studies into atypical and healthy brain development.

## Methods

DeMBA was created by transforming the brain region segmentations from the Allen mouse brain Common Coordinate Framework to each of the developing mouse brain templates. To achieve this, we used Elastix to perform an automated, nonlinear registration. To improve the transformations, we used manually defined corresponding landmarks or regions in temporally adjacent pairs of template volumes. The quality of the transformation was assessed using a set of landmarks defined by a neuroanatomical expert. We then interpolated the transformations, creating models for every postnatal day in between template ages. Lastly, we implemented the atlas in the suite of software of the QUINT workflow[18,30]. In the following, we first describe the reference template and segmentation files, before outlining the methodology for each procedure: 3D-to-3D registration, validation, interpolation, and implementation in software. Lastly, we describe the analysis of calbindin data using DeMBA with the QUINT workflow.

### Reference template and segmentation files

**Allen Common Coordinate Framework reference template and segmentations.** We used the Allen mouse brain Common Coordinate Framework (Allen CCFv3[2]) template at 10 μm isotropic resolution. This template is a population-averaged serial two-photon tomography template based on 1675 subjects (male and female wild-type (C57BL/6 J) and transgenic mice, see Wang and colleagues[2] for details). We further used the 2017 and 2022 editions of the segmentation volume, both at a 10 μm resolution. The two segmentation volumes differ in the amount of annotated brain regions, with the most recent edition having more detailed segmentations in the hippocampus and olfactory bulb. The template and segmentations are available from the Allen Institute archive (https://download.alleninstitute.org/informatics-archive/current-release/mouse_ccf).

To match the resolution and dimensions of the developmental reference templates (see section below), we downsampled the template and segmentation volumes from the Allen Institute to 20 μm and padded the posterior end of the volume with 45 extra voxels. Note that a recent study found that the Allen CCFv3 template display some shrinking in the rostrocaudal dimension (see ref. 36, their Fig. 5), which is not the case for the developmental templates.

**Developmental reference templates.** We used the serial two-photon tomography volumes provided in the supplementary material of the publication by Newmaster and colleagues[13] (male and female Otr-Venus mice on a 129 × C57BL/6 J mixed background). These files were shared as one .tif file for each age (P7, P14, P21, and P28). The original volumes had non-isotropic voxels with a 20 × 20 × 50 μm resolution. We resampled them to isotropic 20 × 20 × 20 μm voxel size using AMIRA (version 6.3.0; https://www.thermofisher.com/no/en/home/electron-microscopy/products/software-em-3d-vis/amira-software.html). We also used the P4 volume shared by Liwang and colleagues[14] (male and female Vip-Cre; Ai14 mice), which was provided as a .tif file with isotropic 20 × 20 × 20 μm voxel size. All volumes were converted

to .nii.gz files for further processing. We further used the 3D developmental common coordinate framework (DevCCFv001) segmentation volume provided by Kronman and colleagues[15,24]. The DevCCFv001 atlas is provided in Allen CCFv3 space at 10 μm resolution, but contains segmentations following a developmental vertebrate ontology[29,37] resembling what was used for the 2D Allen Developing Mouse Brain Atlas.

To optimise the developmental reference templates for successive co-registration and use in software, we implemented several pre-processing steps. First, we removed a slight tilt around the mediolateral axis in templates P7, P14, P21 and P28, ensuring they were as symmetric as possible. Second, we determined the Waxholm Space origin[38] for all the templates and aligned them to the corresponding coordinates in the Allen CCFv3. Next, we standardised the dimensions of the templates as they had a longer rostrocaudal dimension than the Allen CCFv3. To amend this, we cropped some rostrally situated voxels in the developmental reference templates, as these voxels were outside the brain, and their removal did not result in any data loss. However, cropping voxels in the caudal part would affect voxels representing the brain, and we therefore resolved this by adding voxels in the caudal part of the Allen CCFv3 template (see section above). We furthermore adjusted the brightness and sharpness of the templates and adjusted the histograms to the same mean intensity value. The developmental reference templates, including detailed information about each pre-processing step, are available through the EBRAINS research infrastructure[39–43].

### 3D-to-3D registration using elastix

To aid the 3D-to-3D registration, we applied a Contrast Limited Adaptive Histogram Equalisation (CLAHE) filter[44] to enhance the contrast in the templates prior to the registration. To perform the registration for each template pair, we used the SimpleITK Python package (https://simpleelastix.readthedocs.io/) and elastix[23] (https://elastix.lumc.nl/). We performed a three-step (translation, affine and B-spline) registration, with the B-spline registration using pairs of manually defined, corresponding landmarks or regions in the two templates to optimise the registration.

To define corresponding regions across templates, we used a combination of manual and automatic segmentation in ITK-snap[45] (version 3.8.0 and 4.0.2; http://www.itksnap.org/). Defining corresponding regions were only used in cases where a region showed overall poor fit, and the use of a single coordinate to improve it was infeasible, e.g., for the granular cell layer of the dentate gyrus.

Corresponding landmarks were defined using ITK-snap. We experienced that the addition of many manually defined landmarks made the registration overall worse. Even when these landmarks were rigorously defined in terms of anatomical correspondence, they did not necessarily aid the registration when positioned in areas where the automatic procedure already performed well. We therefore opted to add individual landmarks on a need-to-improve basis and evaluating the result of adding individual landmarks after running the registration. As a first step, we ran the automatic registration with no landmarks added, and the result was used as a baseline upon which we gradually refined the registration by adding landmarks in areas where the registration was deemed suboptimal.

We performed an age-to-age registration, where the registration for each template was performed by use of the template closest to its age, i.e., temporal neighbours (Fig. 6). The challenge with this approach was that errors in the initial registration will propagate down through subsequent registrations. To avoid this, we used the transformed template from the previous registration for each registration, as this template would have perfect correspondence with the transformed segmentations. The exception was the P28-P21 registration (Allen CCFv3 template warped to P28). Because of the differences in acquisition between the Allen CCFv3 and the developmental reference

templates, the benefit of not propagating small errors was outweighed by the challenges caused by the relatively dissimilar appearance of the transformed adult template. Thus, the Allen CCFv3 was first registered to the P28 template. Then, the P28 template was registered to the P21 template. The volume resulting from registering the P28 template to P21 was used in the subsequent registration to the P14 template, and so on, until the lowest age (P4) was reached. For each registration, we used the general procedure of gradual refinement by use of landmarks described above; the exact landmarks used, and challenges resolved in each registration, therefore differed across the age groups. Detailed information about each registration, including information about parameters and landmarks or regions used, are available on the EBRAINS Knowledge Graph[46–50]. All the registration codes used here is available on Github (https://github.com/ingvildeb/DeMBA_scripts).

### Validation of the transformations using landmarks

To assess the accuracy of the DeMBA transformation, we used a landmark-based approach. A neuroanatomy expert selected a set of 70 landmarks across the adult mouse brain template (P56), yielding x, y, z coordinates for each landmark. These were documented with a description and coordinates in the Allen CCFv3. Four neuroanatomy experts (including the one who established the landmarks, henceforth called raters) then identified and recorded these landmarks across all DeMBA templates (P4, P7, P14, P21 and P28) using ITK-snap to identify coordinates and noting these down in a spreadsheet. As the group of raters completed this task, it was noted that some of the landmarks were not uniquely identifiable in three planes, and others were not identifiable in themselves (i.e., they were defined by their relation to other landmarks). While such landmarks may be reliably identified *within* a volume, they do not necessarily represent corresponding points *across* volumes. We therefore excluded such landmarks from the validation, based on discussion and agreement among the raters. If any point was not identified by all neuroanatomists in a particular age, it was also excluded from the comparison for that age (this occurred just twice and only in P4). Thus, a final set of 54 landmarks were included in the accuracy analysis. All landmarks are listed with descriptions and coordinates in Supplementary Data 1. The same landmarks were then transformed with the CCF translator, using the DeMBA spatial transformation matrix (hereafter referred to as DeMBA matrix), to corresponding coordinates in the template ages.

The DeMBA matrix coordinates were compared with the median coordinate of the corresponding point from four raters. For each rater, the Euclidean distance from each coordinate to the median coordinate of all other raters was calculated. Since the raters' accuracy was measured against the median coordinate of the 3 other raters, we used the same number of raters when measuring the DeMBA matrix, i.e., the Euclidean distance from each coordinate to the median coordinate for 3 of the raters. This was done for every unique combination of 3 raters, and the result was averaged. For each of the templates (P28, P21, P14, P7 and P4), the distance for all coordinates were averaged across raters and compared to the DeMBA matrix average distance using a two-tailed *t* test. This approach takes advantage of the "wisdom of the crowd" phenomenon when comparing DeMBA to human expertise, as the average output of a group is often more accurate than that of a single individual[51]. When comparing distances, we used the median instead of the mean as this metric is less susceptible to influence by large outliers.

It is worth noting that, as the recording coordinates was a manual process, some of the coordinates with large distances in the raters might represent typographical errors. However, due to the difficulty in retrospectively determining whether a typo occurred and the risk of correcting genuine variability between raters, we decided to leave these coordinates in for our validation. The code for the validation analysis used in this study is available on GitHub (https://github.com/ingvildeb/DeMBA_scripts).

### Interpolation of transformation matrices and application to segmentations

To transform segmentation volumes across ages and create models of the intermediate ages between the template ages (P4, P7, P14, P21, P28 and P56), we used the transformations obtained through Elastix. Specifically, we used the transformation from each temporal neighbour registration to calculate transformation matrices, here referred to as *backwards* transformation matrices, as it transforms data backwards in age. These matrices were then inverted to create the *forwards* transformation matrices. For example, the *backwards* transformation matrix from the P56 to P28 was inverted to generate the *forwards* transformation matrix from P28 to P56. For each intermediate age, we applied the *backwards* and *forwards* matrices to the closest older and younger template ages, which resulted in both template ages being in the space of the intermediate age. We then averaged the two resulting volumes, weighting each according to how close the intermediate age was to the template ages. For example, when generating an interpolated P8 model, both the P7 and P14 templates are transformed into the intermediate P8 space. As P7 is closer in age to P8 than P14, the transformed P7 template is weighted more than the P14 template. The segmentation volumes were only transformed down from the adult template and not vice versa, thus, only the *backwards* transformation matrix was applied to these volumes. We generated 47 models for the intermediate ages, which resulted in a total of 53 template and model volumes.

Lastly, the template ages and interpolated model volumes for every day from P4 to P56 were combined into a compiled 4D volume with time as a continuous dimension. To keep the file size of this compiled volume manageable, we split it into three volumes representing the first (P4–P20), second (P21–P37) and third (P38–P56) subsets of the age range. Detailed information about the files and how to use them is available on the EBRAINS Knowledge Graph[17].

To streamline the transformation and interpolation processes, we created a Python package called *CCF translator*, which allows translation of coordinates and image volumes across atlases. The CCF translator enables a user to provide data sampled from any age and then transform it into a specified target space or age (see Fig. 4 in the main text). This enables direct voxel to voxel comparisons between datasets from different ages. CCF translator is available via GitHub (https://github.com/brainglobe/brainglobe-ccf-translator) and can be installed via pip.

### Implementation in software

To facilitate the use of DeMBA, we implemented the atlas in the suite of software composing the QUINT workflow[18–21]. This includes the QuickNII software[19] for linear registration of section images to 3D reference atlases and VisuAlign[20,30], which allows non-linear refinement of QuickNII registrations. While we used the DeMBA atlas with the Nutil software (v0.8) through the custom atlas function for our study, the DeMBA atlas has now been fully implemented in Nutil (v1.1). We implemented all the DeMBA volumes (53 ages) in a single QuickNII package and added a pop-up window where the user can select which age of the atlas to launch. We also implemented DeMBA in VisuAlign with separate packages for the adult Allen CCFv3 and the developmental segmentation volumes (Allen CCFv3, 2017 edition; Allen CCFv3, 2022 edition[2], DevCCFv001[24], respectively). VisuAlign automatically opens the atlas age to which the data were QuickNII-registered when the user selects a file. The QuickNII and VisuAlign versions with DeMBA implemented can be downloaded through the EBRAINS Knowledge Graph[17].

### Analysing calbindin data using DeMBA with the QUINT workflow

To exemplify the use of DeMBA in the QUINT workflow (https://quint-workflow.readthedocs.io/), we analysed calbindin-stained serial sections from four brains (aged P4, P14, P28 and P56), accessed via the

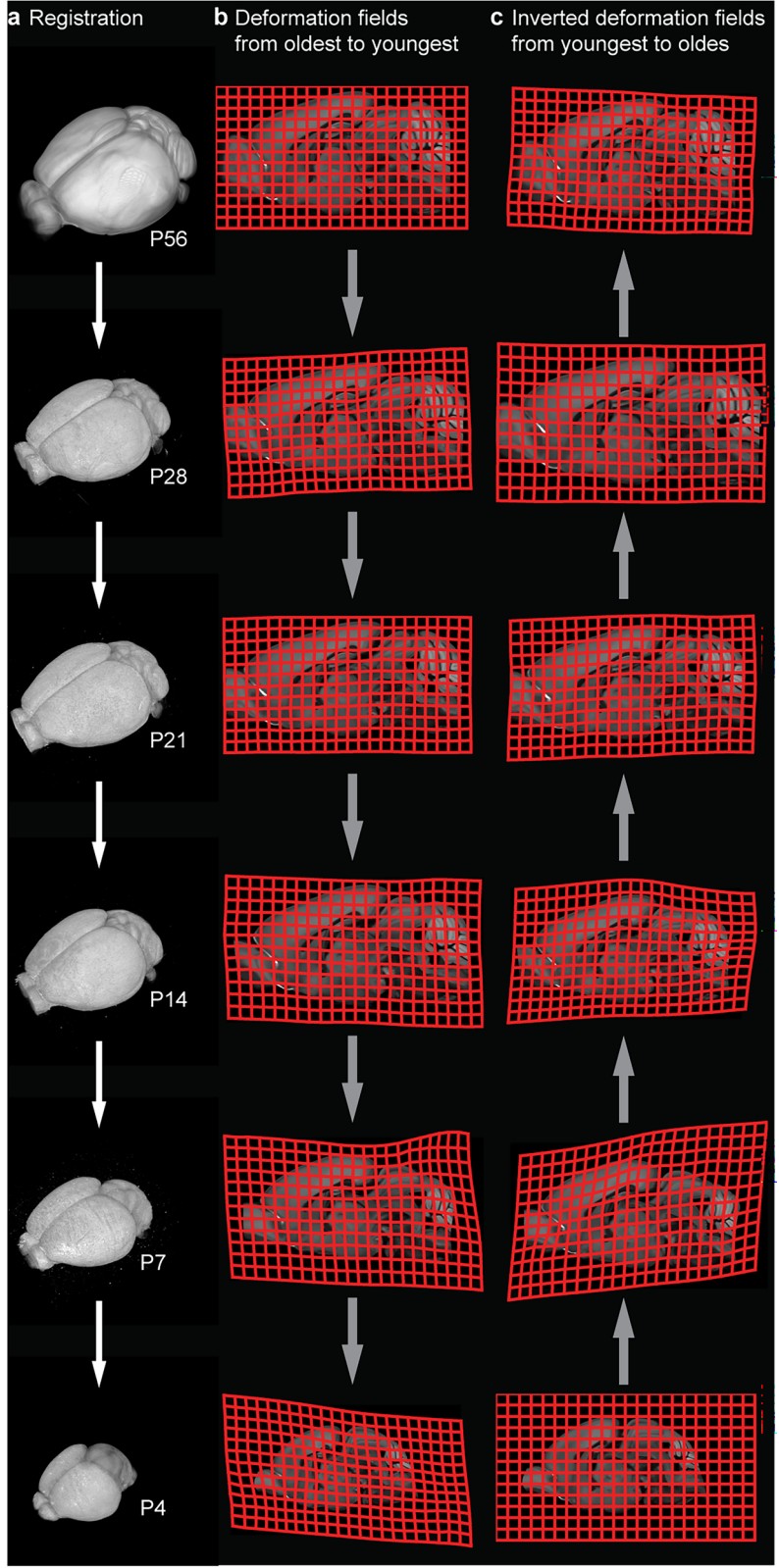

**Fig. 6 | Age-to-age registration of STPT templates from P56 to P4. a** Volume rendering of each STPT template, from P56 (top) to P4 (bottom), which were registered to each other sequentially. **b** Illustration of the deformation fields defined for each (forward) transformation, from oldest to youngest, in a pairwise manner. **c** The deformation fields in (**b**) were inverted to define the (backwards) transformations from youngest to oldest. Averaging and weighing the forward and backwards transformations allowed us to create interpolated models of intermediate ages (see text for details).

Allen Institute in situ hybridisation database[28,29]. The selected series included experiment IDs 100059245 (P4), 100045313 (P14), 100045526 (P28), and 71717640 (P56). Both raw images and corresponding segmented images for each series were downloaded. We registered the images to age-matched DeMBA templates using QuickNII[19] (v2.2; RRID:SCR_016854; https://quicknii.readthedocs.io/) and refined the registration using VisuAlign[20] (v0.9; RRID:SCR_017978; https://visualign.readthedocs.io/). Atlas maps were exported for both adult (Allen CCFv3, 2017 edition) and developmental (DevCCFv001) segmentations and then combined with the segmented images using Nutil[21] (v0.8.0; RRID:SCR_017183; https://nutil.readthedocs.io). The custom atlas feature was used to quantify the calbindin-expressing neurons with DeMBA. We analysed the load of calbindin expression in major brain regions, excluding white matter tracts and ventricles, with the adult and the developmental segmentation.

To create 4D visualisations of calbindin expression, we first reconstructed 3D volumes from each of the spatially registered 2D section datasets. We placed the segmented gene expression images into the empty DeMBA space volumes according to the spatial location assigned to them via QuickNII and VisuAlign. We then filled the gaps between the sections using a K-nearest neighbours' algorithm, which weighted each neighbour by its distance to the empty voxel. This resulted in 3D volumes for each of the ages for which we had data. To interpolate data for the intermediate ages, we used the CCF translator. All the resulting calbindin expression volumes were then concatenated into three 4D volumes representing the same age ranges as in DeMBA (Fig. 5a).

### Reporting summary

Further information on research design is available in the Nature Portfolio Reporting Summary linked to this article.

## Data availability

All data are available via the EBRAINS Knowledge Graph (RRID:SCR_017612; https://www.ebrains.eu/). DeMBA, including developmental reference templates, interpolated models and segmentations, and the spatial registration software incorporating DeMBA (QuickNII v2.2 and VisuAlign v0.9), are available from the dataset below. Nutil version 1.1 is available from the Nutil NITRC homepage (https://www.nitrc.org/projects/nutil). We encourage researchers who use DeMBA to cite the current publication and the dataset listed below, as well as to specify (1) the age of any template(s) used and (2) the version of any segmentation(s) used.

● Carey, H., Ovsthus, M., Kleven, H., Yates, S. C., Csucs, G., Puchades M. A., Bjaalie, J.G., Leergaard, T. B., & Bjerke, I.E. (2024). Developmental mouse brain atlas (DeMBA) with continuous coverage of postnatal day 4–56 (v2) [Data set]. EBRAINS. https://doi.org/10.25493/V3AH-HK7.

In addition, we encourage users to cite the publications describing the data used to create DeMBA, listed below.

● Wang, Q., Ding, S.-L., Li, Y., Royall, J., Feng, D., Lesnar, P., Graddis, N., Naeemi, M., Facer, B., Ho, A., Dolbeare, T., Blanchard, B., Dee, N., Wakeman, W., Hirokawa, K.E., Szafer, A., Sunkin, S.M., Oh, S.W., Bernard, A., … Ng, L. (2020). The Allen Mouse Brain Common Coordinate Framework: A 3D Reference Atlas. *Cell*, *181*(4), 936-953.e20. https://doi.org/10.1016/j.cell.2020.04.007.

● Liwang, J. K., Kronman, F.A., Minteer, J.A., Wu, Y.-T., Vanselow, D.J., Ben-Simon, Y., Taormina, M., Parmaksiz, D., Way, S.W., Zeng, H., Tasic, B., Ng, L., & Kim, Y. (2023). epDevAtlas: Mapping GABAergic cells and microglia in postnatal mouse brains. *bioRxiv*, 2023.11.24.568585. https://doi.org/10.1101/2023.11.24.568585.

● Newmaster, K. T., Nolan, Z. T., Chon, U., Vanselow, D. J., Weit, A. R., Tabbaa, M., Hidema, S., Nishimori, K., Hammock, E. A. D., & Kim, Y. (2020). Quantitative cellular-resolution map of the oxytocin receptor in postnatally developing mouse brains. *Nature Communications*, *11*(1), 1885. https://doi.org/10.1038/s41467-020-15659-1.

The reference data for the template ages are available from the following datasets on the EBRAINS Knowledge Graph:

● Carey, H., Kleven, H., Leergaard, T. B., & Bjerke, I. E. (2024). Population-averaged 3D isotropic serial two-photon tomography reference data for the P4 mouse brain (v1) [Data set]. EBRAINS. https://doi.org/10.25493/VNF6-E92.

● Carey, H., Kleven, H., Qu, H., Leergaard, T.B., & Bjerke, I.E. (2024). Population-averaged 3D isotropic serial two-photon tomography reference data for the P7 mouse brain (v2) [Data set]. EBRAINS. https://doi.org/10.25493/JKBM-608.

● Carey, H., Kleven, H., Qu, H., Leergaard, T.B., & Bjerke, I.E. (2024). Population-averaged 3D isotropic serial two-photon tomography reference data for the P14 mouse brain (v2) [Data set]. EBRAINS. https://doi.org/10.25493/KAHK-14.

● Carey, H., Kleven, H., Qu, H., Leergaard, T.B., & Bjerke, I.E. (2024). Population-averaged 3D isotropic serial two-photon tomography reference data for the P21 mouse brain (v2) [Data set]. EBRAINS. https://doi.org/10.25493/2FMJ-152.

● Carey, H., Kleven, H., Qu, H., Leergaard, T.B., & Bjerke, I.E. (2024). Population-averaged 3D isotropic serial two-photon tomography reference data for the P28 mouse brain (v2) [Data set]. EBRAINS. https://doi.org/10.25493/TG8Z-PER.

All the data related to the transformation between the templates are available from the following datasets:

● Carey, H., Ovsthus, M., Kleven, H., Leergaard, T.B., & Bjerke, I. E. (2024). Allen Mouse Brain CCFv3 segmentations transformed to P4 population-averaged serial two-photon tomography data (v2) [Data set]. EBRAINS. https://doi.org/10.25493/10.25493/4PPR-NVN.

● Ovsthus, M., Carey, H., Kleven, H., Leergaard, T.B., & Bjerke, I.E. (2024). Allen Mouse Brain CCFv3 segmentations transformed to P7 population-averaged serial two-photon tomography data (v2) [Data set]. EBRAINS. https://doi.org/10.25493/CN8M-3Z5.

● Ovsthus, M., Carey, H., Kleven, H., Leergaard, T.B., & Bjerke, I.E. (2024). Allen Mouse Brain CCFv3 segmentations transformed to P14 population-averaged serial two-photon tomography data (v2) [Data set]. EBRAINS. https://doi.org/10.25493/75H6-8VK.

● Ovsthus, M., Carey, H., Kleven, H., Leergaard, T.B., & Bjerke, I.E. (2024). Allen Mouse Brain CCFv3 segmentations transformed to P21 population-averaged serial two-photon tomography data (v2) [Data set]. EBRAINS. https://doi.org/10.25493/WGHE-8HY.

● Ovsthus, M., Carey, H., Kleven, H., Leergaard, T.B., & Bjerke, I.E. (2024). Allen Mouse Brain CCFv3 segmentations transformed to P28 population-averaged serial two-photon tomography data (v2) [Data set]. EBRAINS. https://doi.org/10.25493/6BP7-1X4.

## Code availability

All the code used in this study is available from the DeMBA Github repository (https://github.com/ingvildeb/DeMBA_scripts). The CCF translator is available from the brainglobe-ccf-translator Github repository (https://github.com/brainglobe/brainglobe-ccf-translator).

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

## Acknowledgements

We thank Signý Benediktsdóttir and Sophia Pieschnik for help and advice with data curation through the EBRAINS Knowledge Graph, Hong Qu for expert technical assistance and Adam Tyson for assistance with integrating the CCF translator with BrainGlobe. This work received funding from the European Union's Horizon 2020 Framework Programme for Research and Innovation under the specific grant agreement no. 945539 (Human Brain Project SGA3; J.G.B.); the European Union's Research and Innovation Programme Horizon Europe under Grant Agreement no. 101147319 (EBRAINS 2.0; J.G.B., T.B.L.); and UNIFOR-FRIMED (T.B.L., I.E.B.).

## Author contributions

H.C. and H.K. contributed equally. H.C., H.K and I.E.B. conceived the study with input from J.G.B. and T.B.L. H.C. and I.E.B developed methods and code. H.C., M.Ø. and I.E.B. performed transformations. H.C., M.Ø., I.E.B., H.K., and T.B.L. provided neuroanatomical feedback during registration. H.C., H.K., I.E.B. and S.M. performed expert neuroanatomical landmark identification. I.E.B., S.C.Y., M.A.P. and G.C implemented the atlases in software and contributed to testing and documentation of them. H.C. and I.E.B. prepared use-case data. H.K. and I.E.B. organised all the data for sharing. H.K and IEB created figures. H.C. and I.E.B. draughted the first version of the manuscript. H.C., H.K. and I.E.B. revised the manuscript with input from all authors. I.E.B., T.B.L. and J.G.B. obtained funding. IEB supervised and coordinated the study.

## Competing interests

The authors declare no competing interests.
