## [Transparent Peer Review file · Nature Communications]

DeMBA: A developmental atlas for navigating the mouse brain in space and time

Corresponding Author: Dr Ingvild Bjerke

Version 0:

Reviewer comments:

Reviewer #1

(Remarks to the Author)

This article raises an important issue in the field that many standardized atlases are limited to a few age ranges and don't allow for accurate data registration. The result is that the neuroscience research field narrows in on specific ages or brain regions where registration is valid, but this ultimately limits the study of the brain. This manuscript details the creation of a Developmental Mouse Brain Atlas (DeMBA) which uses the previously published Allen CCFv3 and DevCCF atlas templates at multiple age timepoints (P4, P7, P14, P21, P28, P56) and interpolates new atlases for each remaining postnatal day. The authors show how this new comprehensive atlas can be used to register and analyze multiple 3D experimental datasets including microscopy, MRI, and in situ hybridization gene expression. Gene expression data can also be modeled by interpolation across the aging timepoints. Overall, this is a well-written study that produces a new tool for investigations of postnatal development.

Major concerns: One of the most interesting and novel aspects of DeMBA is the interpolated atlases constructed from the 6 template atlases (P4, P7, P14, P21, P28, P56). However, many of the main figures demonstrating DeMBA's application mainly rely on the template aging timepoints rather than the interpolated timepoints. In Figs. 1-3, P35 is the only interpolated timepoint shown. Fig.5 shows interpolated gene expression only from P9. As these atlas templates are already published and available online, I believe Figs 1 and 5 would be much more interesting and informative if multiple interpolated timepoints were shown instead of the template timepoints. Fig. 5 c-e also shows translation of template gene expression datasets rather than into the interpolated space. Overall, I think the figures should be re-worked to showcase the interpolated atlas timepoints.

Minor concerns: In the paragraph starting at lines 119, it is described how the atlases are validated by expert neuroanatomists, but it is unclear if these neuroanatomists are co-authors of this study or independent, unbiased experts. The author contributions section appears a bit vague listing 'quantitative' or "Qualitative" validation, please clarify either of these sections.

At line 120, it states that 54 landmarks were identified to use as registration points across templates. In the Supplementary Methods, it is stated that first 70 landmarks were used but then some were abandoned. Information about the anatomical landmarks should be listed or included in a supplementary table for reproducibility.

(Remarks on code availability)

Authors have presented and made code available in numerous places including CodeOcean, eBRAINS website, and github repositories.

Reviewer #2

(Remarks to the Author)

DeMBA: a developmental atlas for navigating the mouse brain in space and time

Summary

This manuscript describes a 4D atlas of the developing mouse brain for the first 52 consecutive days of postnatal

development. The atlas is based on open data for 6 time points during development to adult stage. Intermediate days are obtained by pairwise registration and interpolation between these time points. The authors combined 2 common reference atlases, the developmental and adult Allen mouse brain atlas, into this continuous space, providing a reference for mouse brain developmental studies and enabling quantitative analyses across development. Examples are provided for how to register new data and different modalities into this framework. The work is shared in an openly accessible common framework for exploration, analysis and visualisation, assuring easy reusability. The figures are clear and informative. This is an important resource for the community and good fit for publication in Nature Communications.

Minor comments:

1. The data used could potentially be made clearer. From the abstract we get the impression that imaging data had been acquired for all 56 postnatal days. Only at the end of the introduction we learn that it is generated based on brain image templates from 6 time points, and that anatomical images for all other stages stem from interpolation for all days in between these 6 datasets, and segmentation comes from 2 time points. Discussion page 18 “atlas segmentations adapted to age-specific brain morphologies” – which are both interpolations, the anatomies as well as the segmentations.

2. Page 3, second paragraph: the authors mention existing atlases and dive directly into their shortcomings. It would be nice to credit in very short their value and opportunities they provide, before mentioning the gap they have left open so far.

3. Page 9: 54 landmarks are being mentioned, which were drawn by expert neuroanatomists. The authors could maybe add a table to list these 54 landmarks they selected or a figure to show them. Page 9 reads “identified by 4 other expert neuroanatomists” which made me wonder why, but then on page 26 the authors state that it’s 4 including the initial expert. So it should be correct to delete “other” on page 9.

4. Comments on figures:

- Figure 2 is a great overview figure. It could enhance readability to not only call the panels a1, a2 etc, but P4, P7 etc. for easy readability.
- Figure 2a. The plots nicely show the accuracy. The figure caption could maybe mention the average or mean distance as a range P4 to P56?
- Figure 2c nicely shows the accuracy by region. The caption could mention which region(s) showed most variability and which least.
- Page 16, Figure 5 caption reads for white arrowheads to “indicate areas where expression is markedly higher in the adult than P4, black arrowheads indicate areas where expression is markedly lower in P4 than in the adult” → this is slightly redundant, could probably be simplified into one version “black and white arrows indicate...”

5. Comment on videos: The brain appears to grow until ~P19 and then shrink. Part of this is likely due to interindividual variability. It may be good to comment on that, to avoid users of the atlas misusing the data.

6. Page 5 states that “DEMBA was constructed from serial two-photon tomography (STPT) volumes sourced from public datasets [...] hereafter referred to as “templates””. We first understand this as if the atlas were based on a series of single-individual datasets. Maybe make it clearer that these are templates reconstructed from several specimens for 6 time points and these templates have been downloaded from public resources? This information is only provided on page 21, in the Online Methods section.

7. At the end of the manuscript, the authors make citation recommendations, asking authors to cite their current manuscript and their dataset on EBRAINS with versioning number. That’s great, but in addition to this new resource, it would be fair to include the 3 original references: Wang et al. 2020 (P56) (ref 1); Liwang et al. 2023 (P4) (ref12); Newmaster et al. 2020 (P7, P14, P21, P28) (ref 13).

K. Heuer and R. Toro

(Remarks on code availability)

The code is accessible, and coding style looks appropriate and well commented. The code includes tests and test data. We have not attempted to reproduce the results.

Reviewer #3

(Remarks to the Author)

(Remarks on code availability)

The code is accessible, the coding style looks appropriate and the code is well documented. The repository includes tests and test data. I did not attempt to reproduce the results of the paper.

Version 1:

Reviewer comments:

Reviewer #1

(Remarks to the Author)

I appreciate the authors response and their revisions have greatly enhanced the manuscript. I have no further concerns and I believe the manuscript is ready for publication.

(Remarks on code availability)

Reviewer #2

(Remarks to the Author)

The authors addressed all our comments and we recommend publication of the presented manuscript in your journal.
Roberto Toro and Katja Heuer

(Remarks on code availability)

The code is accessible, the coding style looks appropriate and the code is well documented.
The repository includes tests and test data. I did not attempt to reproduce the results of the paper.

Reviewer #3

(Remarks to the Author)

I co-reviewed this manuscript with one of the reviewers who provided the listed reports. This is part of the Nature Communications initiative to facilitate training in peer review and to provide appropriate recognition for Early Career Researchers who co-review manuscripts.KH

(Remarks on code availability)

Reviewer #1 (Remarks to the Author):

Reviewer comment: *“This article raises an important issue in the field that many standardized atlases are limited to a few age ranges and don’t allow for accurate data registration. The result is that the neuroscience research field narrows in on specific ages or brain regions where registration is valid, but this ultimately limits the study of the brain. This manuscript details the creation of a Developmental Mouse Brain Atlas (DeMBA) which uses the previously published Allen CCFv3 and DevCCF atlas templates at multiple age timepoints (P4, P7, P14, P21, P28, P56) and interpolates new atlases for each remaining postnatal day. The authors show how this new comprehensive atlas can be used to register and analyze multiple 3D experimental datasets including microscopy, MRI, and in situ hybridization gene expression. Gene expression data can also be modeled by interpolation across the aging timepoints. Overall, this is a well-written study that produces a new tool for investigations of postnatal development.”*

Response: We thank Reviewer 1 for kind comments and are pleased that the manuscript is perceived as well-written. Below, we provide point-by-point responses to the concerns raised.

Reviewer comment: *“Major concerns: One of the most interesting and novel aspects of DeMBA is the interpolated atlases constructed from the 6 template atlases (P4, P7, P14, P21, P28, P56). However, many of the main figures demonstrating DeMBA’s application mainly rely on the template aging timepoints rather than the interpolated timepoints. In Figs. 1-3, P35 is the only interpolated timepoint shown. Fig.5 shows interpolated gene expression only from P9. As these atlas templates are already published and available online, I believe Figs 1 and 5 would be much more interesting and informative if multiple interpolated timepoints were shown instead of the template timepoints. Fig. 5 c-e also shows translation of template gene expression datasets rather than into the interpolated space. Overall, I think the figures should be re-worked to showcase the interpolated atlas timepoints.”*

Response: We agree with Reviewer 1 that the interpolated templates are one of the most novel aspects of our study and that they were insufficiently highlighted. To address this, we have reworked Figures 1, 3 and 5. Figure 1 now displays 3 interpolated timepoints. We chose to still display two template volumes (P4 and P56) to clearly show the youngest and oldest ages represented in DeMBA, which also highlights the substantial changes in the neuroanatomy across this timespan. For Figure 3, we have updated panels a-b to show an interpolated age

point instead of a template, so that all the examples in this figure shows interpolated ages. We also added panel labels specifying the ages shown. Figure 5 was revised to display interpolated gene expression from two additional interpolated time points (three in total). We also included a more detailed visualization of one of these interpolated ages with a new panel (panel c in the new figure). The remaining panels (now panels d-f) were combined to make room for these expansions. To make it clearer which data are templates/interpolated existing data or interpolated models/interpolated model data, we have separated the two above and below the timeline shown in Figure 1 and 5 and marked the exemplified the ages with full (template) and dashed (interpolated model) circles. Adjustments to figure legends reflecting these changes have been made throughout. We hope that the updated figures better showcase the interpolated timepoints.

Reviewer comment: *Minor concerns: In the paragraph starting at lines 119, it is described how the atlases are validated by expert neuroanatomists, but it's unclear if these neuroanatomists are co-authors of this study or independent, unbiased experts. The author contributions section appears a bit vague listing 'quantitative' or 'Qualitative' validation, please clarify either of these sections.*

Response: We agree that the contributions of the experts should be highlighted and have adjusted the author contributions to reflect which authors contributed expert neuroanatomical landmark identification (line 619-620). Additionally, we have adjusted the contributions to more explicitly define what was meant by qualitative validation.

Reviewer comment: *At line 120, it states that 54 landmarks were identified to use as registration points across templates. In the Supplementary Methods, its stated that first 70 landmarks were used but then some were abandoned. Information about the anatomical landmarks should be listed or included in a supplementary table for reproducibility.*

Response: We agree that information about these landmarks is important for reproducibility, and we thank the reviewer for drawing our attention to them. We have included a table (Supplementary file 1) that lists all information regarding these landmarks and added a reference to this at the line mentioned by the reviewer (now line 125).

Reviewer comment: *Authors have presented and made code available in numerous place including CodeOcean, eBRAINS website, and github repositories.*

Response: We thank the reviewer for acknowledging our efforts to make the code findable and reusable.

Reviewer #2 (Remarks to the Author):

Reviewer comment: *This manuscript describes a 4D atlas of the developing mouse brain for the first 52 consecutive days of postnatal development. The atlas is based on open data for 6 time points during development to adult stage. Intermediate days are obtained by pairwise registration and interpolation between these time points. The authors combined 2 common reference atlases, the developmental and adult Allen mouse brain atlas, into this continuous space, providing a reference for mouse brain developmental studies and enabling quantitative analyses across development. Examples are provided for how to register new data and different modalities into this framework. The work is shared in an openly accessible common framework for exploration, analysis and visualisation, assuring easy reusability.*

The figures are clear and informative. This is an important resource for the community and good fit for publication in Nature Communications.

Response: We thank Reviewer 2 for kind comments and are pleased that our atlas is found to be an important resource.

Reviewer comment: *Minor comments:*

1. The data used could potentially be made clearer. From the abstract we get the impression that imaging data had been acquired for all 56 postnatal days. Only at the end of the introduction we learn that it is generated based on brain image templates from 6 time points, and that anatomical images for all other stages stem from interpolation for all days in between these 6 datasets, and segmentation comes from 2 time points. Discussion page 18 “atlas segmentations adapted to age-specific brain morphologies” – which are both interpolations, the anatomies as well as the segmentations.

Response: We appreciate this observation and have adjusted the abstract to now highlight that our resource is a combination of existing data and interpolated timepoints (lines 23 to 25).

Reviewer comment: 2. Page 3, second paragraph: the authors mention existing atlases and dive directly into their shortcomings. It would be nice to credit in very short their value and opportunities they provide, before mentioning the gap they have left open so far.

Response: We agree that it is important to credit currently available resources and make clear the impact they have had. We have adjusted the manuscript to reflect this (lines 44 to 49).

Reviewer comment: 3. Page 9: 54 landmarks are being mentioned, which were drawn by expert neuroanatomists. The authors could maybe add a table to list these 54 landmarks they selected or a figure to show them. Page 9 reads “identified by 4 other expert neuroanatomists” which made me wonder why, but then on page 26 the authors state that it’s 4 including the initial expert. So it should be correct to delete “other” on page 9.

Response: We thank the reviewer for catching this error and have adjusted line 127 to reflect the correct number of neuroanatomists. In addition, we have added a table (Supplementary file 1, line 125) to fully document the landmarks.

Reviewer comment: 4. Comments on figures:

- Figure 2 is a great overview figure. It could enhance readability to not only call the panels a1, a2 etc, but P4, P7 etc. for easy readability.

Response: We agree with the reviewer that this enhances readability and have added panel labels indicating the ages on the left side of the figure.

Reviewer comment: - Figure 2a. The plots nicely show the accuracy. The figure caption could maybe mention the average or mean distance as a range P4 to P56?

Response: We have updated the caption of figure 2a to include the range of mean error values across the ages (line 146-148).

Reviewer comment: - Figure 2c nicely shows the accuracy by region. The caption could mention which region(s) showed most variability and which least.

Response: We agree this is interesting and have updated the caption to figure 2c (line 154-155) to mention which regions showed the highest error on average. We did not see any trends across ages with some regions having particularly low error.

Reviewer comment: - Page 16, Figure 5 caption reads for white arrowheads to “indicate areas where expression is markedly higher in the adult than P4, black arrowheads indicate areas where expression is markedly lower in P4 than in the adult” → this is slightly redundant, could probably be simplified into one version “black and white arrows indicate...”

Response: We agree and have adjusted this sentence in the Figure 5 legend to be less verbose (line 247-249).

Reviewer comment: 5. Comment on videos: The brain appears to grow until ~P19 and then shrink. Part of this is likely due to interindividual variability. It may be good to comment on that, to avoid users of the atlas misusing the data.

Response: We thank the reviewer for this observation. We believe that this is in large part due to the methodological differences in the studies generating the developmental (up to P28) and adult template. We have expanded our discussion (lines 275 to 279) to reflect this limitation of our resource.

Reviewer comment: 6. Page 5 states that “DEMBA was constructed from serial two-photon tomography (STPT) volumes sourced from public datasets [...] hereafter referred to as “templates””. We first understand this as if the atlas were based on a series of single-individual datasets. Maybe make it clearer that these are templates reconstructed from several specimens for 6 time points and these templates have been downloaded from public resources? This information is only provided on page 21, in the Online Methods section.

Response: We have addressed this by adding a sentence on lines 87 to 88 to indicate that the templates are population averages. In addition, we have added specific information regarding how many animals are included in each template.

Reviewer comment: 7. At the end of the manuscript, the authors make citation recommendations, asking authors to cite their current manuscript and their dataset on EBRAINS with versioning number. That’s great, but in addition to this new resource, it would be fair to include the 3 original references: Wang et al. 2020 (P56) (ref 1); Liwang et al. 2023 (P4) (ref12); Newmaster et al. 2020 (P7, P14, P21, P28) (ref 13).

Response: An exciting development in the field of neuroinformatics is the degree to which different publications have been able to build on the findings of others. This has been made possible by the enthusiasm with which researchers in our field have engaged in data sharing.

A challenge that comes with this development is the proper way in which to properly assign credit. We fully agree with the reviewer that it is appropriate to encourage users of DeMBA to also cite the publications which made the creation of DeMBA possible, and we have adjusted our citation recommendations to reflect this (lines 555 to 571).

Reviewer comment: *Reviewer #2 (Remarks on code availability):*

The code is accessible, and coding style looks appropriate and well commented. The code includes tests and test data. We have not attempted to reproduce the results.

Response: We thank Reviewer 2 for their kind comments.

Reviewer #3:

Reviewer comment: *(Remarks on code availability):*

The code is accessible, the coding style looks appropriate and the code is well documented. The repository includes tests and test data. I did not attempt to reproduce the results of the paper.

Response: We thank Reviewer 3 for recognising our effort to make the code accessible and reusable. We hope that others may find it of use.

Reviewer #1 (Remarks to the Author):

I appreciate the authors response and their revisions have greatly enhanced the manuscript. I have no further concerns and I believe the manuscript is ready for publication.

Author response: We are happy to hear that the reviewer find our manuscript ready for publication.

Reviewer #2 (Remarks to the Author):

The authors addressed all our comments and we recommend publication of the presented manuscript in your journal.
Roberto Toro and Katja Heuer

Reviewer #2 (Remarks on code availability):

The code is accessible, the coding style looks appropriate and the code is well documented. The repository includes tests and test data. I did not attempt to reproduce the results of the paper.

Author response: We are pleased to hear that the reviewers recommend our manuscript for publication.

Reviewer #3 (Remarks to the Author):

I co-reviewed this manuscript with one of the reviewers who provided the listed reports. This is part of the Nature Communications initiative to facilitate training in peer review and to provide appropriate recognition for Early Career Researchers who co-review manuscripts.KH